# High-Intensity Focused Ultrasound Decreases Subcutaneous Fat Tissue Thickness by Increasing Apoptosis and Autophagy

**DOI:** 10.3390/biom13020392

**Published:** 2023-02-18

**Authors:** Kyung-A Byun, Hyun Jun Park, Seyeon Oh, Sosorburam Batsukh, Hye Jin Sun, Taehui Kim, Sunggeun Kim, Donghwan Kang, Kuk Hui Son, Kyunghee Byun

**Affiliations:** 1Department of Anatomy & Cell Biology, College of Medicine, Gachon University, Incheon 21936, Republic of Korea; 2Functional Cellular Networks Laboratory, Lee Gil Ya Cancer and Diabetes Institute, Gachon University of Medicine, Incheon 21999, Republic of Korea; 3Maylin Anti-Aging Center Apgujeong, Seoul 06005, Republic of Korea; 4Jeisys Medical Inc., Seoul 08501, Republic of Korea; 5Department of Thoracic and Cardiovascular Surgery, Gachon University Gil Medical Center, Gachon University, Incheon 21565, Republic of Korea

**Keywords:** high-intensity focused ultrasound, subcutaneous adipose tissue, apoptosis, autophagy

## Abstract

High-intensity focused ultrasound (HIFU) leads to decreased subcutaneous adipose tissue (SAT) thickness via heat-induced adipocyte necrosis. Heat can induce adipocyte apoptosis and autophagy, and it is known that nuclear or mitochondrial p53 is involved in apoptosis and autophagy. However, whether HIFU leads to apoptosis or autophagy is unclear. We evaluated whether HIFU decreases SAT thickness via p53-related apoptosis or autophagy in high-fat diet (HFD)-fed animals. The expression of nuclear and mitochondrial p53 was increased by HIFU. HIFU also led to decreased expression of BCL2/BCL-xL (an antiapoptotic signal), increased expression of BAX/BAK (an apoptotic signal), increased levels of cleaved caspase 3/9, and increased numbers of apoptotic cells as evaluated by TUNEL assay. Furthermore, HIFU led to increased levels of ATG5, BECN1, and LC3II/LC3I, and decreased levels of p62, a marker of increased autophagy. The thickness of SAT was decreased by HIFU. In conclusion, HIFU led to nuclear and mitochondrial p53 expression, which led to apoptosis and autophagy, and eventually decreased SAT thickness in HFD-fed animals.

## 1. Introduction

High-intensity focused ultrasound (HIFU) has been frequently used for body sculpting without surgery via ablation of undesirable adipose tissue [1]. HIFU generates heat above 58 °C only in the targeted area, without affecting adjacent tissue [2]. The generated heat induces mechanical disruption of the adipocyte membrane or coagulative necrosis, which leads to immediate cell death [2].

Histologic examination of adipose tissue subjected to HIFU has demonstrated the presence of pyknotic nuclei and vacuoles in the extracellular material [3]. Necrotic cellular debris and lipids are removed by macrophages [1]. It has been considered that the main cell death process of HIFU in decreasing subcutaneous adipose tissue (SAT) thickness is heat-induced adipocyte necrosis. Extreme temperatures that lead to thermal ablation, such as those above 60 °C, induce necrosis; however, hyperthermia or heat stress (40–45 °C) induces apoptosis [4,5,6,7].

A recent study has shown that heat stress induces p53-dependent apoptosis [8]. Upon various cellular stresses, nuclear p53 increases the transcription of a family of genes including Puma, Noxa, Bax, and Bid, and thus leads to apoptosis [9]. However, cytoplasmic p53 directly upregulates the mitochondrial death pathway in a transcription-independent manner [10].

Cytoplasmic p53, translocated into mitochondria, interacts with Bcl-2 family members, which can be either antiapoptotic (BCL-xL, BCL2, MCL-1) or proapoptotic (PUMA, BAX, BAK) [11,12]. The binding of p53 with BCL-2 or BCL-xL causes BAK and BAX to be released from BCL2 or BCL-xL; thus, homo-oligomerization of BAX and BAX is increased, which induces the generation of permeable pores on the outer membranes of mitochondria [13,14,15,16,17]. Through these pores, cytochrome C is released, which activates caspase 9 and caspase 3, and ultimately induces apoptosis [18,19]. In addition to apoptosis, p53 is involved in the regulation of the autophagy machinery [20].

When cellular stress is controllable, autophagy is initiated for cell survival. However, autophagy can induce cell death when cellular stress is extreme [21]. Downregulation of BCL2 and BCL-xL via binding with p53 leads to the release of beclin-1 (BECN1), which ultimately activates autophagy [22,23]. However, cytoplasmic p53 induces BECN1 degradation and ultimately inhibits autophagy [12].

Under conditions of cellular stress, p53 induces the activation of autophagy-related genes (ATGs), such as ATG5 and ATG7 [24,25]. Heat treatment also increases the expression of BECN1 and Light Chain 3 (LC3) II, which are autophagosome formation markers [26].

Even though heat-induced p53 activation leads to various types of cell death, such as apoptosis or autophagy, it has not been fully revealed whether HIFU induces apoptosis or autophagy as well as cell necrosis. Thus, we evaluated whether HIFU leads to p53-induced autophagy and apoptosis in the SAT of high-fat diet (HFD)-fed animals.

## 2. Materials and Methods

### 2.1. Animal Model

Animal experiments were conducted by the guidelines of the Institutional Animal Care and Use Committee. In addition, the study was approved by the Center of Animal Care and Use Ethical Board of Gachon University (approval number: LCDI-2021-0135).

Sprague-Dawley rats (220 ± 20 g, male, 8 weeks old) were obtained from Orient Bio (Sungnam, Republic of Korea) and acclimatized for one week. The rats were housed in cages with a 12 h light/dark cycle under a controlled temperature (22 °C) and relative humidity (50%), and up to two rats were housed in one cage.

After one week of acclimatization, the animals were randomly divided into groups. The normal-fat diet (NFD) group consumed a NFD for 21 days. The HFD and HFD/HIFU groups consumed 45% HFD for 7 days; after HIFU was applied, the rats consumed the diet at 6 h HFD/18 h NFD. All rats had free access to diet and water and their body weight was measured once a week for 21 days during the experiment.

### 2.2. HIFU System and Applied to Rats

A HIFU system (LinearZ, Jeisys Medical Inc., Seoul, Republic of Korea) was used in this study [27]. A 7 MHz transducer was used, and the DOT mode was applied with 7 MHz frequency, 3.0 mm focal depths, and 0.6 J energy. The rats in the HFD/HIFU groups were treated with HIFU on the dorsal SAT, and the skin and SAT were collected after 1, 7, and 14 days of HIFU application.

### 2.3. In Vitro Model

The mouse pre-adipocyte, 3T3-L1 MBX (ATCC, Manassas, VA, USA) was cultured in growth medium (Dulbecco’s Modified Eagle’s Medium (Hyclone, UT, USA), high glucose, containing 10% fetal bovine serum (Gibco, Grand Island, NY, USA) and 1% penicillin-streptomycin (Welgene, Daegu, Republic of Korea) at 37 °C in a humidified atmosphere of 5% CO_2_.

When the 3T3-L1 MBX cell density was confluence to differentiate into mature adipocytes, the culture medium was replaced with MDI media (growth medium supplemented including 0.5 mM 3-isobutyl-1-methylxanthine (Sigma-Aldrich, ST. Louis, MO, USA), 0.5 µM dexamethasone (Sigma-Aldrich), and 5 µg/mL insulin (Sigma-Aldrich)). After MDI induction for 2 days, the differential media were exchanged for growth medium supplemented with 5 μg/mL insulin (Insulin medium). The medium was changed every 2 days. After 4 days of exposure to the Insulin medium, the media were replaced with growth media for another 2 days [28,29].

The mature adipocytes were treated with or without pifithrin-α hydrobromide as a p53 inhibitor (PFT-α, 30umol/mL; MedChemExpress, Monmouth Junction, JN, USA) for 12 h [30], and then HIFU (Dot mode, 7MHz frequency, and 0.6 J energy) applied or not applied to the mature adipocytes and cultured for 2 days. Then, the samples were harvested.

### 2.4. Nucleus, Cytosol, and Mitochondria Isolation and Protein Preparation

Nuclear and cytoplasmic isolation (Thermo Scientific, Waltham, MA, USA), and mitochondrial isolation (Thermo Scientific), were performed according to the manufacturer’s instructions. Briefly, 100 mg of frozen tissue was homogenized with a Bioprep-24R (Allsheng, Hangzhou, China) using the kit buffer.

Additionally, protein extraction was performed according to the manufacturer’s instructions. Fifty milligrams of frozen tissue or cells were homogenized with a Bioprep-24R (Allsheng) using an EzRIPA lysis kit (ATTO, Tokyo, Japan) according to the manufacturer’s instructions.

### 2.5. Western Blot

Thirty micrograms of protein were loaded into 8–12% polyacrylamide gels for separation by electrophoresis (Criterion System, Bio-Rad Laboratories, Inc., Hercules, CA, USA). The proteins were transferred onto PVDF membranes (Millipore, Burlington, MA, USA), and the membranes were incubated overnight with the primary antibodies listed in Appendix A. The proteins were visualized using peroxidase-conjugated secondary antibodies (Vector Laboratories Inc., Newark, CA, USA) and enhanced chemiluminescence substrate (Cytiva, Vancouver, Canada) on a digital acquisition system (Bio-Rad). Individual protein expression values were quantified using ImageJ software (NIH, Bethesda, MD, USA) and normalized to those of beta-actin to control for differences in protein loading. The values for a single blot are expressed relative to the NFD or pre-adipocyte group’s mean.

### 2.6. Preparation of Paraffin-Embedded Skin and SAT Tissue Sections

The skin tissues attached to SAT were fixed with cold 4% paraformaldehyde (Sigma-Aldrich). The fixed tissue samples were washed for 1 h for embedding. Paraffin blocks of skin tissues attached to SAT were then created using a tissue processor (Thermo Fisher Scientific). The paraffin-embedded blocks were sectioned at 7 µm using a microtome (Leica, Wetzlar, Germany) and dried at 60 °C for 24 h to keep them attached to the coated slides.

### 2.7. TdT-Mediated dUTP Nick-End Labeling Assay

TUNEL assay (TransDetect^®^ In Situ Fluorescein TUNEL Apoptosis Detection Kit; TransGen Biotech, Beijing, China) was performed according to the manufacturer’s instructions. Briefly, the sectioned slides were incubated with xylene and in a descending ethanol series (100%, 95%, 80%, and 70%) for deparaffinization and washed with distilled water. Then, the slides were rinsed with phosphate-buffered saline (PBS) and incubated with permeabilization solution for 5 min at room temperature. After incubation with labeling solution for 1 h at 37 °C, the labeled slides were rinsed with permeabilization solution 3 times. The washed slides were incubated with 4’,6-Diamidino-2-Phenylindole, dihydrochloride (DAPI), and mounted with VECTASHIELD (Vector Laboratories Inc.) mount.

### 2.8. Hematoxylin and Eosin Staining

To measure the thickness of SAT, the skin tissues attached to SAT were stained with hematoxylin and eosin. Briefly, deparaffinized and rehydrated tissue slides were immersed in hematoxylin solution (KPNT, Cheongju, Republic of Korea) and washed with tap water for 3 min. The tissue slides were incubated with ammonia water for 30 s, immersed in eosin solution (KPNT) for 1 min, rinsed with running water, and dehydrated. The coverslips were mounted using DPX solution (Sigma-Aldrich), and the slides were visualized under an optical microscope (BX53M; Olympus, Japan). The thickness of the SAT was determined by randomly collecting 10 SAT images using ImageJ software (NIH).

### 2.9. Real-Time Quantitative Reverse Transcription Polymerase Chain Reaction (qRT-PCR)

Total RNA was extracted from cells using RNAiso (Takara, Kusatsu, Japan), and cDNA was synthesized from RNA via a PrimeScipt 1^st^ strand Synthesis Kit (TaKaRa). The procedures were conducted according to the respective manufacturer’s protocols. Expression analysis of the reported genes was performed by real-time PCR with SYBR Premix (TaKaRa). Data were analyzed via the relative Ct (2−ΔΔCt) method and were expressed as a fold change of gene expression level compared with the respective pre-adipocyte group. The sequences of the primer pairs used in this study are listed in Appendix A.

### 2.10. Statistical Analysis

The data were validated from at least three replicates of each experiment and are presented as the means ± standard deviations. In this study, the Kruskal–Wallis test was used for comparisons of five groups, and the Mann–Whitney U test was used for post hoc analysis using SPSS v.22 (IBM Corporation; Armonk, NY, USA). Statistical significance is represented as follows: *, HFD group vs. NFD group or normal mature adipocyte group vs. normal pre adipocyte group; $, vs. HFD group or normal mature adipocyte group.

## 3. Results

### 3.1. HIFU Decreased the Thickness of the SAT

We applied HIFU to the dorsal SAT of HFD-fed animals (the HFD/HIFU group) and harvested SAT at 1, 7, and 14 days after HIFU. The tissue was compared with tissue from NFD-fed animals without HIFU (the NFD group) and tissue from HFD-fed animals without HIFU (the HFD group) at 21 days after initiation of the HFD or NFD (14 days after HIFU in the HFD/HIFU animals). In the HFD group and HFD/HIFU group, 45% of HFD was fed to the animals for 24 h during the first 7 days. The amount of the HFD was reduced for 6 h during the 14 days after HIFU was applied in the HFD/HIFU group. We believed that HFD feeding during the whole experimental period might lead to extreme obesity, which could mask the ability of HIFU to decrease SAT thickness (Figure 1A).

The body weights of the HFD/HIFU group and HFD group were significantly higher than those of the NFD group at 7, 14, and 21 days after HIFU application. However, there were no significant differences in body weight between the HFD/HIFU group and the HFD group at 7, 14, and 21 days after HIFU application (Figure 1B).

The thickness of SAT in the HFD group was significantly higher than that of the NFD group (Figure 1C,D). The thickness of SAT in the HFD/HIFU group at 1 day was not significantly different from that of the HFD group; however, the thickness of SAT in the HFD/HIFU group at 7 and 14 days was significantly lower than that of the HFD group.

### 3.2. HIFU Increased the Expression of Nuclear and Mitochondrial p53 and Decreased the Expression of Cytoplasmic p53 in the SAT

We isolated the nuclear, cytoplasmic, and mitochondrial fractions of tissue samples to evaluate whether HIFU induced upregulation of p53 in the nucleus and mitochondria. The expression of nuclear p53 in the SAT was not significantly different between the NFD and HFD groups. However, the expression of nuclear p53 was significantly higher in the HFD/HIFU group at 1, 7, and 14 days than in the HFD group (Figure 2A,B).

The expression of mitochondrial p53 in the SAT did not significantly differ between the NFD and HFD groups. However, the expression of mitochondrial p53 was significantly higher in the HFD/HIFU group at 1, 7, and 14 days than in the HFD group (Figure 2A,C).

The expression of cytoplasmic p53 was significantly higher in the HFD group than in the NFD group. However, the expression of cytoplasmic p53 was significantly lower in the HFD/HIFU group than in the HFD group at 1, 7, and 14 days (Figure 2A,D).

### 3.3. HIFU Decreased BCL2/BCL-xL Expression and Increased BAX/BAK Expression and Cytochrome C Release

The expression of mitochondrial BCL2 and BCL-xL in the HFD group was significantly higher than that of the NFD group. The expression of mitochondrial BCL2 and BCL-xL in the HFD/HIFU group at 1, 7, and 14 days after HIFU was significantly lower than that of the HFD group (Figure 3A–C).

The expression of BAX in the HFD group was not significantly different from that of the NFD group. However, it was significantly higher than that of the HFD/HIFU group at 1, 7, and 14 days after HIFU (Figure 3A,D). 

The expression of BAK in the HFD group was not significantly different from that of the NFD group. It was also not significantly different from that of the HFD/HIFU group at 1 day after HIFU; however, it was significantly higher than that of the HFD/HIFU group at 7 and 14 days after HIFU (Figure 3A,E).

The expression of cytochrome C in the cytoplasm was significantly lower in the HFD group than in the NFD group and significantly higher in the HFD/HIFU group at 1, 7, and 14 days after HIFU than in the HFD group (Figure 3F,G).

### 3.4. HIFU Decreased BCL2/BCL-xL Expression and Increased BAX/BAK Expression and Cytochrome C Release via Modulation of p53

To evaluate whether HIFU decreased BCL2/BCL-xL expression and increased BAX/BAK expression via modulating p53, we designed an in vitro study. After inducing the differentiation of 3T3-L1 cells into mature adipocytes, HIFU was applied. In addition, inhibition of p53 in the mature adipocyte was performed with PFT-α, and those cells were also irradiated by HIFU (Appendix A, Appendix A).

In the mature adipocyte, HIFU increased the expression of p53. However, the expression of p53 was not changed by HIFU after the inhibition of p53 in the mature adipocyte (Appendix A).

The expression of BCL2/BCL-xL was decreased, in the mature adipocyte, by HIFU. The expression of BCL2/BCL-xL was higher in the p53-inhibited mature adipocyte than in the normal mature adipocyte. In the p53-inhibited mature adipocyte, HIFU did not decrease the expression of BCL2/BCL-xL (Appendix A).

The expression of BAX was increased, in the mature adipocyte, by HIFU. The expression of BAX was not different between normal and p53-inhibited mature adipocytes. In the p53-inhibited mature adipocyte, HIFU did not increase expression of BAX (Appendix A).

The expression of BAK was increased, in the mature adipocyte, by HIFU. The expression of BAX was lower in the p53-inhibited mature adipocyte than in normal adipocyte. In the p53-inhibited mature adipocyte, HIFU did not increase the expression of BAK (Appendix A).

The expression of cytochrome C was increased, in the mature adipocyte, by HIFU. The expression of cytochrome C in the p53 inhibited mature adipocyte was lower than the normal mature adipocyte. In the p53-inhibited mature adipocyte, HIFU did not increase expression of cytochrome C (Appendix A).

### 3.5. HIFU Induced Caspase 3/9 Activation and Apoptosis in the SAT

The ratio of cleaved caspase 3 and total caspase 3 (cleaved caspase 3/caspase 3 ratio) was significantly lower in the HFD group than in the NFD group. The cleaved caspase 3/caspase 3 ratio in the HFD/HIFU group at 1, 7, and 14 days after HIFU was significantly higher than that of the HFD group (Figure 4A,B).

The cleaved caspase 9/caspase 9 ratio was significantly lower in the HFD group than in the NFD group. The cleaved caspase 9/caspase 9 ratio in the HFD/HIFU group at 1, 7, and 14 days after HIFU was significantly higher than that of the HFD group (Figure 4A,C).

The number of apoptotic cells evaluated with the TUNEL assay in the HFD group was not significantly different from that of the NFD group. However, the number of apoptotic cells was significantly higher in the HFD/HIFU group at 1, 7, and 14 days after HIFU than in the HFD group (Figure 4D,E).

### 3.6. HIFU Induced p53 Change Involved in Caspase 3/9 Activation

The cleaved caspase 3/caspase 3 and cleaved caspase 9/caspase 9 ratios were increased by HIFU in the mature adipocyte. The cleaved caspase 3/caspase 3 and cleaved caspase 9/caspase 9 ratios were lower in the p53-inhibited mature adipocyte than the normal mature adipocyte. In the p53-inhibited mature adipocyte, HIFU did not increase the expression of the cleaved caspase 3/caspase 3 and cleaved caspase 9/caspase 9 ratios (Appendix A).

### 3.7. HIFU Increased Autophagy in the SAT

Increased BECN1 and LC3II/LC3I ratios have been widely used as autophagy markers [31]. In addition, decreased expression of p62 suggests increased autophagy, since activation of autophagy induces a decrease in p62 expression [32].

The expression of ATG5 in the HFD group was significantly lower than that of the NFD group. It was also significantly lower than that of the HFD/HIFU group at 7 and 14 days after HIFU. However, it was not significantly different from that of the HFD/HIFU group at 1 day after HIFU (Figure 5A,B).

The expression of BECN1 in the HFD group was significantly lower than that of the NFD group. The expression of BECN1 in the HFD/HIFU group at 1, 7, and 14 days was significantly higher than that of the HFD group (Figure 5A,C).

The expression of p62 in the HFD group was significantly higher than that of the NFD group. The expression of p62 in the HFD/HIFU group at 1, 7, and 14 days was significantly lower than that of the HFD group (Figure 5A,D).

The expression of LC3II/LC3I in the HFD group was significantly lower than that of the NFD group. The expression of LC3II/LC3I in the HFD/HIFU group at 1, 7, and 14 days was significantly higher than that of the HFD group (Figure 5A,E).

### 3.8. HIFU Increased Autophagy via Modulation of p53

The expressions of ATG5 and BECN1 were increased, in the mature adipocyte, by HIFU. Those expressions were lower in the p53-inhibited mature adipocyte than in the normal adipocyte. In the p53-inhibited adipocyte, HIFU did not increase the expression of ATG5 or BECN1 (Appendix A).

The expression of p62 was decreased, in the mature adipocyte, by HIFU. The expression of p62 was higher in the p53-inhibited mature adipocyte than in the normal adipocyte. In the p53-inhibited mature adipocyte, HIFU did not decrease p62 (Appendix A).

The ratio of LC3II/LC3I was increased by HIFU. The ratio of LC3II/LC3I was lower in the p53-inhibited mature adipocyte than in normal adipocyte. In the p53-inhibited mature adipocyte, HIFU did not increase the ratio of LC3II/LC3I (Appendix A).

The notch1 signal pathway is also known to trigger autophagy [33]. Notch1 is upregulated by p53, and Notch1 is decreased by p53 silencing [34]. HIFU increased Notch1 in the mature adipocyte. The expression of Notch1 was lower in the p53-inhibited mature adipocyte than in the normal adipocyte. HIFU did not increase Notch1 in the p53-inhibited mature adipocyte. It seemed that HIFU increased Notch1, which triggers autophagy via activation of p53 (Appendix A).

## 4. Discussion

Our study showed that HIFU decreased the thickness of SAT in HFD-fed animals and increased mitochondrial and nuclear p53 expression, apoptosis, and autophagy. Various cell death processes are involved in maintaining homeostasis by removing unnecessary or dysfunctional cells. Cell death can be categorized as programmed cell death (PCD) or non-PCD depending on whether the death is a physiological or pathological process [35]. PCD includes apoptosis and autophagy, while non-PCD includes necrosis [36].

While necrosis induces inflammation, apoptosis does not. Even if apoptotic cells undergo secondary necrosis, which induces leakage of cellular content, the apoptotic cells are held in an anti-inflammatory state and remain immunologically silent [37,38,39].

HIFU leads to coagulation necrosis of adipocytes, and necrotic adipocytes secrete chemotactic factors that induce mild inflammation via chemotaxis of macrophages and phagocytosis of necrotic debris [1]. However, our study shows that HIFU also leads to increased apoptosis and autophagy.

We hypothesized that heat-induced increases in the expression of mitochondrial or nuclear p53 caused by HIFU would induce apoptosis and autophagy. It is well known that p53 interacts with BCL2 and BCL-xL in the mitochondrial outer membrane. Both BCL2 and BCL-xL are also involved in the control of autophagy and apoptosis [22,23]. In our study, the increase in mitochondrial p53 expression was accompanied by decreases in BCL2/BCL-xL expression, increases in BAX/BAK expression, and an increase in the number of apoptotic cells in the SAT, as evaluated by TUNEL assay. Moreover, HIFU decreased expression of BCL2/BCL-xL in the mature adipocyte, and those decreasing effects disappeared upon p53 inhibition. HIFU also increased the expression of BAX/BAK in the mature adipocyte; however, those increasing effects disappeared upon inhibition of p53. These findings suggest that increased expression of mitochondrial p53 might increase apoptosis of adipocytes.

Nuclear p53 is known to increase the levels of ATG5 and BECN1, which are components of the autophagic machinery. Autophagy is initiated by the formation of a preinitiation complex that consists of ULK1, ATG13, and ATG101 [40]. Then, an initiation complex forms that consist of BECN1, ATG14, and vacuolar protein sorting-associated protein 15 (Vps15) [40]. Activation of the initiation complex requires disruption of the binding between BECN1 and BCL2 [41]. The initiation complex forms a phagophore, and the phagophore transforms into a double-membrane autophagosome structure through an elongation reaction. During the elongation process, LC3-I (the cytosolic form) changes into LC3-II (the autophagosome-bound form) through conjugation with phosphatidylethanolamine (PE) [40]. For the generation of LC3-II-PE, a ubiquitin-like protein conjugation system involving proteins such as ATG3/4/5/7/10/12/16 and E2 ligase is needed [41]. LC3II binds to ubiquitinated proteins via the adaptor protein p62, which mediates autophagy [42].

In our study, HIFU increased nuclear p53 expression, and this effect was accompanied by increased expression of the autophagy markers ATG5, BECN1, and LC3II/LC3I in the SAT. The findings suggest that an increase in nuclear p53 expression leads to increased autophagy. It is known that cytoplasmic p53 inhibits autophagy, and, in our study, cytoplasmic p53 expression was decreased by HIFU. In the mature adipocyte, HIFU increased the expression of ATG5, BECN1, and LC3II/LC3I. However, those increasing effects of HIFU did not appear in the p53-inhibited mature adipocyte. Notch1 expression, which is known to increase autophagy [33], was also increased by HIFU. Upon inhibition of p53, HIFU did not increase Notch1. Since p53 increased Notch1 in the keratinocyte [43], our results suggested that HIFU increased Notch1 via p53.

By increasing apoptosis and autophagy, HIFU decreased the thickness of SAT. We used HFD-fed animals to mimic obese patients undergoing HIFU. HFD feeding increased SAT thickness, while HIFU decreased it. The animals were fed the HFD even after HIFU was applied, until tissues were harvested. The body weight of the animals was higher at the time of tissue harvest than at the initiation of HFD feeding in both the HFD and HFD/HIFU groups. Moreover, there was no difference in body weight between the HFD and HFD/HIFU groups. Even though body weight was not decreased, the thickness of the SAT where HIFU was applied was decreased. We believe that these effects were caused by apoptosis and autophagy. Since apoptosis and autophagy were maintained at increased levels until 14 days after HIFU administration, the thickness of the SAT also continuously decreased over time even while HFD feeding continued. If the decrease in the thickness of SAT caused by HIFU had been induced only by immediate necrosis, the thickness of the SAT might not have decreased over time. Thus, we propose that HIFU may effectively decrease SAT thickness by increasing apoptosis and autophagy, as well as by inducing necrosis.

In conclusion, HIFU led to nuclear and mitochondrial p53 expression, which led to apoptosis and autophagy, and eventually decreased SAT thickness in HFD-fed animals.

## Figures and Tables

**Figure 1 biomolecules-13-00392-f001:**
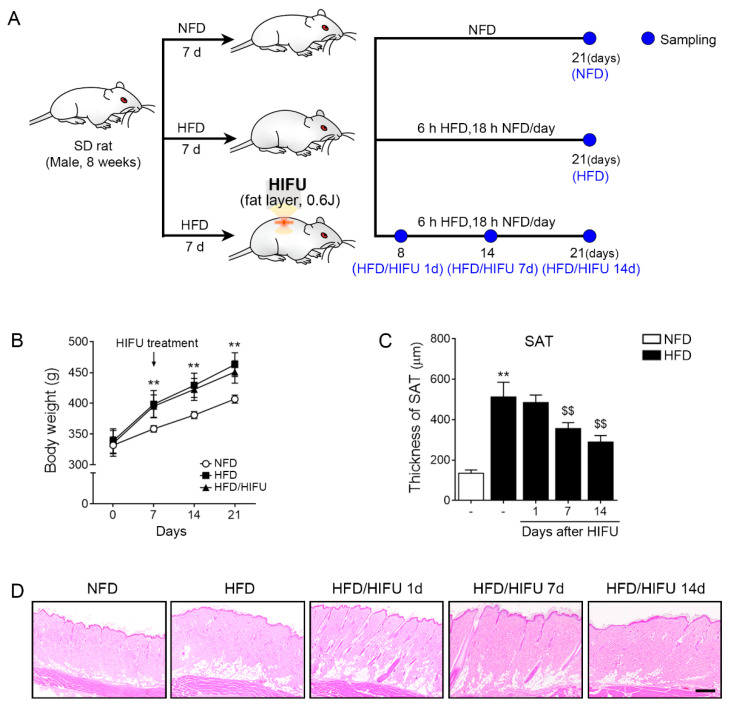
Regulation of SAT thickness by HIFU treatment. (**A**) Schematic diagram of the animal study. (**B**) Body weight changes were analyzed. (**C**,**D**) The thickness of SAT was analyzed by hematoxylin and eosin staining. The thickness of SAT was decreased by HIFU treatment. The data are presented as the mean ± standard deviation; **, *p* < 0.01 HFD vs. NFD; $$, *p* < 0.01 vs. HFD (Mann–Whitney U test). d, days; h, hours; HFD, high-fat diet; HIFU, high-intensity focused ultrasound; NFD, normal-fat diet; SAT, subcutaneous adipose tissue; SD, Sprague–Dawley.

**Figure 2 biomolecules-13-00392-f002:**
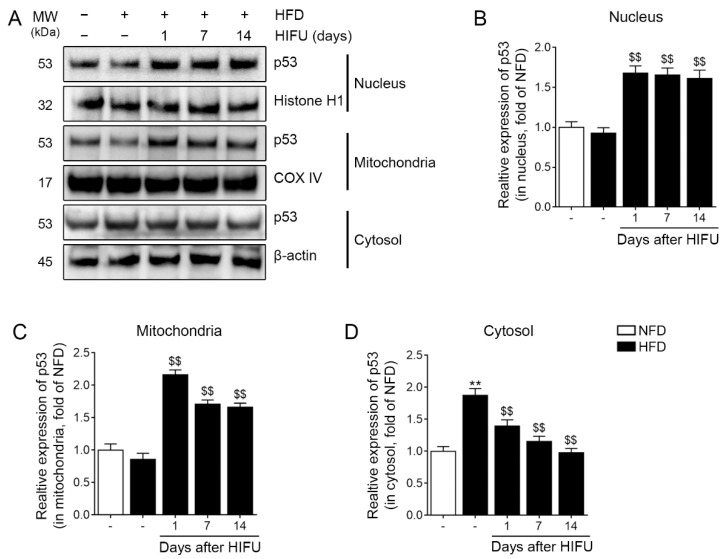
Regulation of nuclear, mitochondrial, and cytoplasmic p53 in SAT by HIFU treatment. (**A**) The protein expression levels of nuclear, mitochondrial, and cytoplasmic p53 were analyzed by western blot. (**B**–**D**) The quantitative graphs represent the results of the western blot analysis. The protein expression of nuclear p53 (**B**) and mitochondrial p53 (**C**) was increased by HIFU treatment. The protein expression of cytoplasmic p53 (**D**) was decreased by HIFU treatment. The data are presented as the mean ± standard deviation; **, *p* < 0.01 HFD vs. NFD; $$, *p* < 0.01 vs. HFD (Mann–Whitney U test). COX IV, cytochrome c oxidase subunit 4; HFD, high-fat diet; HIFU, high-intensity focused ultrasound; NFD, normal-fat diet; SAT, subcutaneous adipose tissue.

**Figure 3 biomolecules-13-00392-f003:**
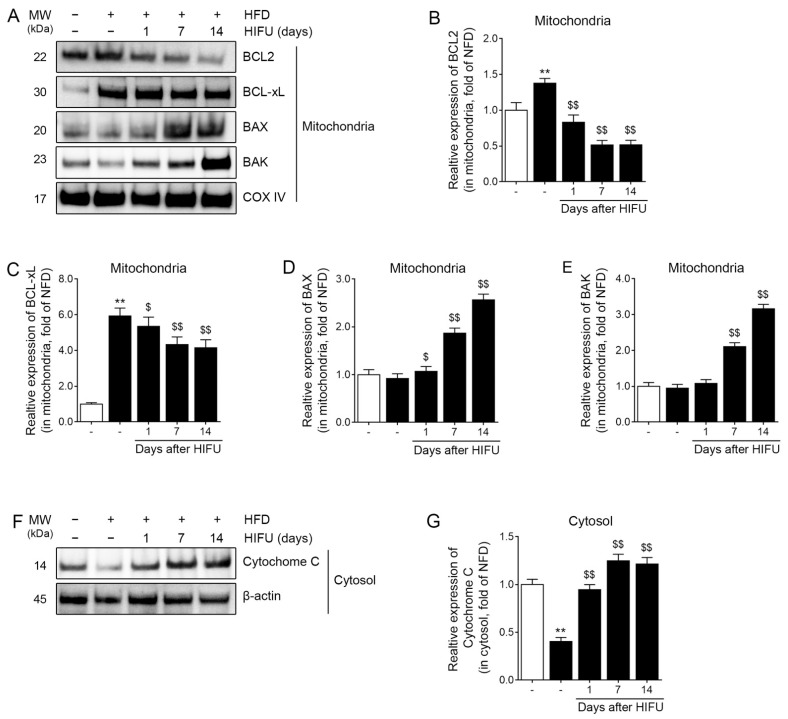
Regulation of the Bcl-2 family in SAT mitochondria and cytochrome C in SAT cytoplasm by HIFU treatment. (**A**) The protein expression levels of Bcl-2 family members were analyzed by western blot. (**B**–**E**) The quantitative graphs represent the results of the western blot analysis. The protein expression levels of BCL2 (**B**) and BCL-xL (**C**) were decreased by HIFU treatment. The protein expression levels of BAX (**D**) and BAK (**E**) were increased by HIFU treatment. (**F**) The protein expression of cytochrome C was analyzed by western blot. (**G**) The quantitative graphs represent the results of the western blot analysis. The protein expression of cytochrome C was increased by HIFU treatment. The data are presented as the mean ± standard deviation; **, *p* < 0.01 HFD vs. NFD; $, *p* < 0.05 or $$, *p* < 0.01 vs. HFD (Mann–Whitney U test). BAK, Bcl-2 homologous antagonist/killer; BAX, bcl-2-like protein 4; BCL2, B-cell lymphoma 2; BCL-xL, B-cell lymphoma-extra large; COX IV, cytochrome c oxidase subunit 4; HFD, high-fat diet; HIFU, high-intensity focused ultrasound; NFD, normal-fat diet; SAT, subcutaneous adipose tissue.

**Figure 4 biomolecules-13-00392-f004:**
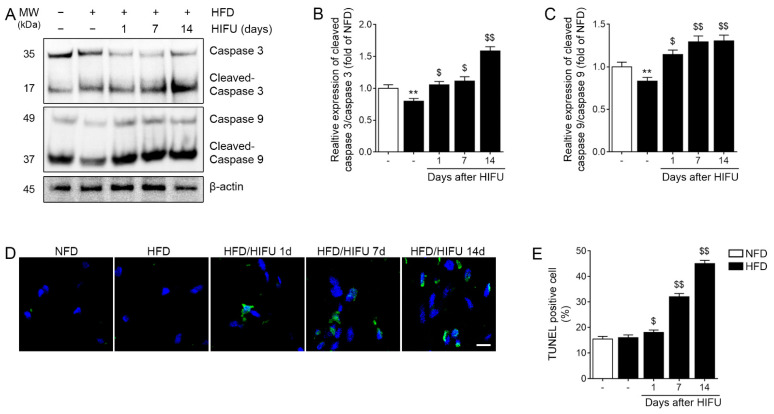
Regulation of apoptosis in SAT by HIFU treatment. (**A**) The protein expression levels of caspase 3 and caspase 9 were analyzed by western blot. (**B**,**C**) The quantitative graphs represent the results of the western blot analysis. The protein expression ratios of cleaved caspase 3/caspase 3 (**B**) and cleaved caspase 9/caspase 9 (**C**) were increased by HIFU treatment. (**D**,**E**) The number of apoptotic cells was analyzed by TUNEL assay. The number of apoptotic cells was increased by HIFU treatment. The data are presented as the mean ± standard deviation; **, *p* < 0.01 HFD vs. NFD; $, *p* < 0.05 or $$, *p* < 0.01 vs. HFD (Mann–Whitney U test). HFD, high-fat diet; HIFU, high-intensity focused ultrasound; NFD, normal-fat diet; SAT, subcutaneous adipose tissue; TUNEL, TdT-mediated dUTP Nick-End Labeling.

**Figure 5 biomolecules-13-00392-f005:**
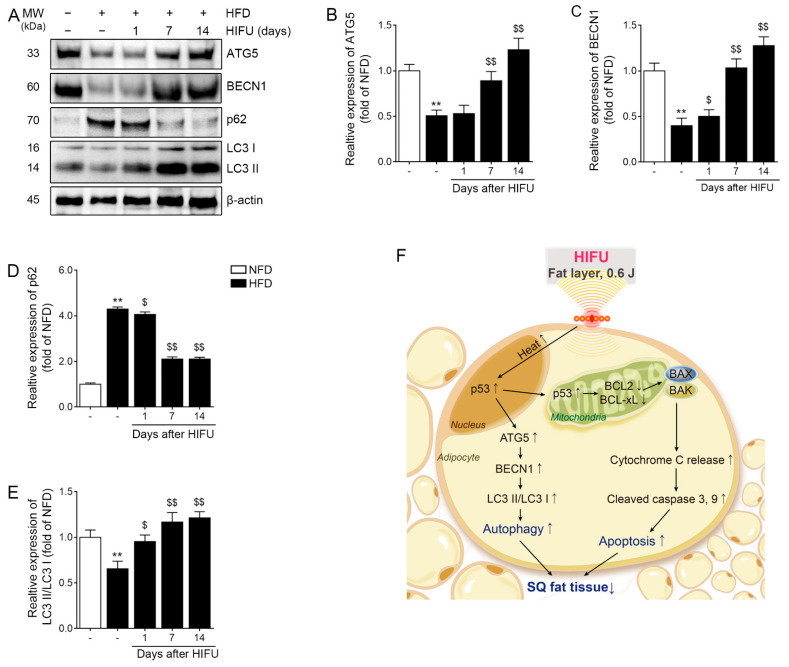
Regulation of autophagy in SAT after HIFU treatment. (**A**) The protein expression levels of autophagy-related markers were analyzed by western blot. (**B**–**E**) The quantitative graphs represent the results of the western blot analysis. The protein expression of ATG5 (**B**), BECN1 (**C**), and p62 (**D**), as well as the ratio of LC3II to LC3I (**E**), were regulated by HIFU treatment. (**F**) Summary of this study. The data are presented as the mean ± standard deviation; **, *p* < 0.01 HFD vs. NFD; $, *p* < 0.05 or $$, *p* < 0.01 vs. HFD (Mann–Whitney U test). ATG5, autophagy related 5; BECN1, beclin-1; HFD, high-fat diet; HIFU, high-intensity focused ultrasound; LC3, microtubule-associated protein 1A/1B-light chain 3; NFD, normal-fat diet; SAT, subcutaneous adipose tissue.

## Data Availability

All data are contained within the article.

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
