# Peer review of "High-Intensity Focused Ultrasound Decreases Subcutaneous Fat Tissue Thickness by Increasing Apoptosis and Autophagy"

_biomolecules, 2023, doi:10.3390/biom13020392_

Round 1

Reviewer 1 Report (Previous Reviewer 1)

line 277

The new added in vitro model is enough to fullfill the conclusion.

"In the mature adipocyte, HIFU decreased the expression of p53." should be changed to "In the mature adipocyte, HIFU increased the expression of p53.".

Author Response

Response to Reviewer 1 Comments

The new added in vitro model is enough to fullfill the conclusion.

  • Point 1: "In the mature adipocyte, HIFU decreased the expression of p53." should be changed to "In the mature adipocyte, HIFU increased the expression of p53.".
  • Response 1: Thank you for comment. We revised manuscript.

Result, Page 8, Line 257 in revised manuscript

In the mature adipocyte, HIFU increased the expression of p53.

Reviewer 2 Report (Previous Reviewer 2)

accepted as it is 

Author Response

Thank you for your review.

This manuscript is a resubmission of an earlier submission. The following is a list of the peer review reports and author responses from that submission.

Round 1

Reviewer 1 Report

This paper have a big gap between p53 and apoptosis/ autophagy. In Figure 2, almost all location of p53 only has <1.5 changes between control and HIFU treatment. The following assay about all the apoptosis or autophagy markers in Figure 3.4.5 cannot conclude that all were real mediated by p53. The subcutaneous skin contains many tumor supressor genes likes TAp63 or NOTCH1 may also involve in apoptosis or autophagy.  It's should use the p53 inhibitor or siRNA in vivo or in vitro system to prove that the apoptosis or autophagy by HIFU is real mediated by p53.

Other points:

1. No N=? in all the figures.

2. In Figure 4D,  it should show the TUNEL only. The merged pic is not easy to find the red spots. 

Reviewer 2 Report

This is a very interesting article. well-structured , that studies the use of High-intensity focused ultrasound (HIFU) which is a method of generating temperature in a sensitive area without affecting other areas. this technique which affects the P53 protein as well as the process of apoptosis:

- we know that the P53 protein is responsible for controlling cell division: can this technique cause cell malformation?

- The activity of apoptosis is stimulated by this technique, so is it possible for hyperactivity and consequently an elimination of healthy cells?

- This technique can be a technique to repair mutations at the level of P53 and contribute to the fight against cancer.

A very long conclusion, can you make it as short as possible?